# Host–Pathogen Interactions during Shiga Toxin-Producing *Escherichia coli* Adherence and Colonization in the Bovine Gut: A Comprehensive Review

**DOI:** 10.3390/microorganisms12102009

**Published:** 2024-10-03

**Authors:** Lekshmi K. Edison, Indira T. Kudva, Subhashinie Kariyawasam

**Affiliations:** 1Department of Comparative, Diagnostic, and Population Medicine, College of Veterinary Medicine, University of Florida, Gainesville, FL 32610, USA; edison.le@ufl.edu; 2Food Safety and Enteric Pathogens Research Unit, National Animal Disease Center, Agricultural Research Service, United States Department of Agriculture, Ames, IA 50010, USA; indira.kudva@usda.gov

**Keywords:** zoonotic transmission, food safety, public health, virulence factors

## Abstract

Shiga toxin-producing *Escherichia coli* (STEC) is a significant public health threat due to its ability to cause severe gastrointestinal diseases in humans, ranging from diarrhea to life-threatening conditions such as hemorrhagic colitis and hemolytic uremic syndrome (HUS). As the primary reservoir of STEC, cattle play a crucial role in its transmission through contaminated food and water, posing a considerable risk to human health. This comprehensive review explores host–pathogen interactions during STEC colonization of the bovine gut, focusing on the role of gut microbiota in modulating these interactions and influencing disease outcomes. We integrated findings from published transcriptomics, proteomics, and genomics studies to provide a thorough understanding of how STEC adheres to and colonizes the bovine gastrointestinal tract. The insights from this review offer potential avenues for the development of novel preventative and therapeutic strategies aimed at controlling STEC colonization in cattle, thereby reducing the risk of zoonotic transmission.

## 1. Introduction

The bovine gut is a complex ecosystem that plays a crucial role in the health and well-being of cattle [1]. Understanding the complex dynamics of host–pathogen interactions during Shiga toxin-producing *Escherichia coli* (STEC) colonization of the bovine gut is pivotal for understanding the epidemiology and pathogenesis of this important foodborne pathogen. While there are more than 470 STEC serogroups, serogroup O157 and 6 other non-O157 serotypes popularly known as the “Big Six”, namely O26, O45, O103, O111, O121, and O145, are considered major foodborne pathogens because of their ability to induce a spectrum of severe gastrointestinal illnesses in humans upon foodborne transmission. These illnesses range from relatively mild diarrhea to life-threatening diseases, such as hemorrhagic colitis and hemolytic uremic syndrome (HUS) [2]. Cattle asymptotically carry these bacteria within the gastrointestinal tract and have been identified as a principal reservoir for STEC. This asymptomatic carriage is a double-edged sword; while it spares cattle from illness, it allows for continuous shedding of these pathogens into the environment. This shedding facilitates zoonotic transmission of STEC to humans, primarily through the consumption of contaminated food and water, underscoring the agricultural and environmental aspects of STEC management [3].

According to the Centers for Disease Control and Prevention (CDC), STEC remains a significant public health concern and causes an estimated 265,000 illnesses each year in the United States alone, leading to approximately 3600 hospitalizations and 30 deaths [4]. Globally, the World Health Organization (WHO) highlights that STEC infections are a leading cause of diarrhea, which is a major cause of morbidity and mortality in children under five years of age in low-income countries [5]. In particular, STEC O157:H7, the most well-characterized serotype, has been responsible for numerous outbreaks associated with contaminated food and food products, including leafy greens, raw milk, and undercooked meat [6]. The economic impact is also significant, with costs associated with healthcare, lost productivity, and food recalls running into millions of US dollars annually [7].

The study of host–pathogen interactions in the context of STEC colonization in cattle is essential for several reasons. First, it sheds light on the mechanisms by which STEC establishes and maintains colonization within the bovine gut, which is fundamental to its persistence and spread within cattle. Understanding these mechanisms can help identify intervention strategies to reduce STEC shedding in cattle, thereby decreasing food and water contamination and the risk of human exposure and infection. Second, exploring the host response to STEC colonization provides insights into the factors contributing to the asymptomatic carrier state in cattle, which is crucial to understanding STEC epidemiology and implementing control strategies. Thirdly, elucidating the complex interplay between STEC, bovine hosts, and gut microbiota opens avenues for developing novel preventative and therapeutic approaches [8]. Understanding host–pathogen interactions at the molecular level can also contribute to developing rapid diagnostic tools and targeted treatments for STEC infections in humans. It is paramount to advance our knowledge of STEC pathogenesis, improve public health interventions, and enhance food safety. The STEC colonization of the bovine gut can lead to a range of health problems, including foodborne illnesses in humans [9,10,11,12]. This review aims to provide a comprehensive overview of the current state of knowledge on the host–pathogen interaction during STEC colonization of the bovine gut, including the key factors that influence host–pathogen interactions, its effects on the human host, and identifying areas where further research is needed.

## 2. STEC Colonization of the Bovine Gut

### 2.1. STEC Prevalence in the Bovine Gut

STEC is a frequent colonizer of the bovine gut, with young calves being particularly susceptible to colonization and frequent shedding compared to adult cattle [13]. Early exposure to STEC via horizontal or vertical transmission contributes to its high prevalence among cattle worldwide [12,14]. Studies have indicated that significant fecal shedding in young calves is influenced by factors such as age and gut microflora composition [10,15,16]. Shiga toxin (Stx) variants, particularly Stx2, correlate with younger calves, highlighting age-related differences in STEC colonization and shedding dynamics [17]. Beef calves shed more STEC during their first six months of age, decreasing as they mature. This shedding pattern also correlates with a lower diversity of gut microbiota in younger animals, which increases as cattle mature [10]. In addition to age-related shedding patterns, factors such as breed, sex, and weight gain also influence STEC shedding in cattle [11].

Among various STEC serotypes, O157:H7 is one of the most studied. It has been implicated in numerous outbreaks of foodborne illnesses worldwide, leading to severe symptoms, such as bloody diarrhea and hemolytic uremic syndrome (HUS) [18]. In addition to STEC O157:H7, several other non-O157 STEC serogroups, including O26, O45, O103, O111, O121, and O145, are also considered major foodborne pathogens, with cattle serving as reservoirs and posing a risk of contamination of food and water sources, subsequently causing disease in humans [19,20]. Table 1 lists the major STEC serogroups in the bovine gut, key virulence factors, and types of illness inflicted in humans. The prevalence of STEC in the bovine gut varies depending on factors such as geographic location, farming practices, and environmental conditions [21,22].

### 2.2. Modes of Transmission from Bovines to Humans and the Potential Risks to Human Health

STEC transmission from cattle to humans occurs via various pathways (Figure 1). The primary mode of transmission from bovine sources to humans is consuming contaminated raw or undercooked milk and meat contaminated with cattle feces during farming and production practices [33,34]. The infectious dose of STEC is low, approximately 100 bacteria, making person-to-person transmission possible and leading to secondary cases due to contact with infected individuals [35]. Direct animal-to-human contact is another crucial pathway for STEC transmission when handling cattle. Visiting farms or venues where the public interacts with farm animals can increase the risk of infection, emphasizing the importance of maintaining hygienic practices in these environments [36]. Waterborne transmission has also been reported, highlighting the need for safe water practices to prevent STEC outbreaks [6].

The risks associated with STEC shedding from bovine sources are substantial and can lead to severe health complications in humans. Symptoms of STEC infection include abdominal cramps, diarrhea, and, in more severe cases, hemorrhagic colitis and HUS. HUS is a life-threatening condition characterized by acute renal failure, hemolytic anemia, and thrombocytopenia, and up to 10% of patients with STEC develop this syndrome [37]. Children under ten years of age are particularly vulnerable to serious STEC infections, with approximately 15% of the children with STEC diarrhea progressing to HUS [38].

## 3. Bovine Host Factors Influencing STEC Colonization

### 3.1. Characteristics of Bovine Gastrointestinal Tract

The unique characteristics of the bovine gastrointestinal tract (GIT), especially at the rectoanal junction (RAJ), play a crucial role in STEC colonization. The RAJ, situated at the terminal end of the ruminant GIT, transitions from columnar epithelial cells in the distal colon to stratified squamous epithelial cells towards the anus. This site is particularly conducive to STEC colonization and persistence and plays a major role in the super-shedding of STEC in cattle feces. Dense lymphoid follicles with different cell types, such as follicle-associated epithelium (FAE) and RAJ squamous epithelium (RSE), play distinct roles in bacterial adherence. STEC promotes adherence to RSE via a specialized adherence strategy that is not absent in commensal bacteria [39,40]. STEC can colonize various sites within the bovine GIT beyond the RAJ. Other parts of the intestine, such as the ileum, jejunum, cecum, colon, and rectum, offer favorable microenvironments for STEC adherence and persistence, emphasizing the ability of bacteria to navigate and exploit different niches within the bovine GIT for survival and multiplication [41].

The pH levels throughout the bovine GIT create favorable regions for the survival and proliferation of STEC. The rumen has a relatively neutral pH, which is favorable to diverse microbiota, and serves as an important reservoir for STEC. However, as the digesta containing STEC move through the GIT, the pH levels fluctuate, creating varying acidic conditions, particularly in the abomasum and small intestine, before STEC stabilizes in the large intestine. These fluctuations in pH provide niches that STEC can uniquely exploit because of its ability to tolerate a wide range of pH [42]. Unlike many other bacteria that cannot withstand lower pH levels, STEC is well adapted to survive in relatively high acidic environments and effectively colonize downstream to the rumen, especially the colon. This acid resistance allows STEC to bypass harsh conditions in the stomach and effectively colonize regions such as the colon, where the pH is higher and more stable. STEC can attach to the intestinal epithelium in the colon, establishing colonization sites and enabling persistence and potential transmission through fecal shedding [43].

### 3.2. Immune Responses and Other Host Factors That Influence STEC Colonization of the Bovine Gut

STEC has evolved mechanisms to evade immune detection and establish colonization in cattle. Studies have shown that calves infected with STEC O157 strains produce antibodies against O157 lipopolysaccharide (LPS) but may suppress cellular immune responses, especially when infected with Stx-producing strains. This suppression is significant because it allows the pathogen to persist in the host and continue to be shed in feces. Stx targets bovine lymphocytes, particularly CD8+ T cells and B cells, such as intraepithelial lymphocytes (IELs), located between epithelial cells adjacent to the basement membrane in the intestines of cattle. The toxin binds to receptors on Gb3/CD77-positive lymphocytes in the early activation stages, which inhibits lymphocyte proliferation. This suppression affects the immune system’s ability to mount an effective cellular response against the pathogen, and the inflammatory response is less pronounced, facilitating the persistence of STEC in the gut [12,44].

STEC effector proteins such as NleE and NleB can inhibit the activation of NF-κB, a central regulator of inflammation. NleE modifies the host ubiquitination machinery and prevents the degradation of IκB, an inhibitor of NF-κB, thereby maintaining NF-κB in an inactive state. In contrast, NleB is a glycosylated death domain-containing protein involved in NF-κB signaling, thereby blocking the NF-κB signaling pathway. By keeping NF-κB inactive, these effectors inhibit the expression of pro-inflammatory cytokines such as IL-2, TNF-α, and IFN-α, dampening the host immune response and allowing the bacteria to evade detection and destruction [23,45]. The Mitogen-Activated Protein Kinase (MAPK) pathway is crucial for producing inflammatory cytokines, the key signaling molecules of the host immune response. T3SS effector proteins, such as NleC, NleD, and OspF, can inhibit the MAPK pathway by interfering with its signaling components, thereby blocking the activation of transcription factors, including AP-1, responsible for cytokine production. This inhibition weakens the host inflammatory response, making it harder for the immune system to detect and eliminate bacteria, thus facilitating bacterial survival and colonization. Both mechanisms do not induce lymphocyte death. Instead, they prevent the proliferation and response of lymphocytes to antigens by blocking the activation of their response to mitogens, the substances that stimulate cell division [46,47,48,49]. This blockage indicates that even if the immune system recognizes the pathogen, it cannot effectively activate the cells needed to fight the infection. Thus, the lymphocytes remain viable but are functionally impaired, increasing host susceptibility to pathogen infection and colonization. Furthermore, the ability of STEC to adhere to the gut epithelium through fimbriae and other adhesins facilitates intimate interactions promoting colonization [12].

Mucosal-associated mechanisms are believed to facilitate the adherence of STEC to the intestinal surface. Various receptors on the mucosal epithelium or in the mucus bind to bacterial adherence factors, including fimbriae, intimin, and other adhesins. The number and affinity of these receptors differ between neonatal and adult animals, leading to age-related differences in susceptibility to STEC intestinal colonization and subsequent excretion, which are higher in neonatal animals [50]. Additionally, factors such as attaching and effacing lesions formed by STEC alter the gut environment, promoting STEC colonization [9]. The diet also has a significant impact on STEC intestinal colonization. Cattle fed forage-based diets showed more shedding than those fed grain-rich diets because volatile fatty acid (VFA) concentrations in grain cattle feed can inhibit the adherence of STEC to the mucosal surface. Younger cattle, especially pre-ruminant calves, are more susceptible to STEC colonization because of their immature immune system, lower levels of specific antibodies, and mucosal surface receptors with more affinity. Pre-weaned calves also have lower VFA concentrations and higher lactate levels in their gut, creating a more favorable environment for STEC colonization [50].

## 4. Role of STEC Virulence Factors in Bovine Colonization

STEC utilizes a complex array of virulence factors, including Stx, adhesins, autotransporters, Type III secretion system effectors, fimbriae, and pili proteins, to colonize the bovine gut successfully while overcoming host defenses. Table 2 summarizes the key aspects of STEC virulence factors, including their roles in colonization. The interplay between STEC virulence factors during bovine colonization is a coordinated process that allows bacteria to establish infection without causing disease, evade host defenses, and persist within the gut environment [51].

### 4.1. Initial Adherence through Adhesins and Fimbriae/Pili

Hair-like structures such as fimbriae and pili on the surface of STEC play crucial roles in bacterial adherence, colonization, and interaction with host cells. These structures are composed of protein subunits and can be categorized into several types, including Type 1 Fimbriae, F9 Fimbriae, Curli Fimbriae, Long Polar Fimbriae (LPF), and *E. coli* YcbQ Laminin-Binding Fimbriae (ELF), based on their function and the genes that encode them. These structures serve as the primary factors for the initial attachment of STEC to the intestinal epithelium leading to colonization [52]. Fimbriae and pili bind to specific receptors on the surface of the bovine intestinal epithelial cells, typically glycoprotein or glycolipid present on the mucosal surfaces. The tip of the fimbriae or pili contains specialized proteins called adhesins, such as IrgA homolog adhesin (Iha) and STEC autoagglutinating adhesin (Saa), which recognize and bind to host receptors with high affinity [53,54,55]. Intimin, encoded by the eae gene of STEC, is another key adhesin that enables the formation of attaching and effacing (A/E) lesions on cattle intestinal epithelial cells. It is a well-studied non-fimbrial adhesin that plays a critical role in tight adhesion to host cells after initial contact mediated by fimbriae/pili [52]. This type of attachment enabled by adhesins, along with fimbriae and pili, is crucial for establishing a foothold in the gut and overcoming mechanical defense mechanisms such as mucus flow [51]. Different types of fimbriae and pili recognize different host receptors, contributing to tissue specificity and tropism. This specificity determines the sites within the gastrointestinal tract where STEC can effectively colonize [51,52,53,56].

### 4.2. Manipulation of Host Cells via Type III Secretion System (T3SS) Effectors

Once STEC is attached, it releases an effector protein called Tir (translocated intimin receptor) into the host gut epithelial cell membrane through T3SS. Once inside the host cell membrane, Tir acts as a strong receptor for intimin and creates a stable, intimate attachment between the bacterium and host cell, anchoring bacteria firmly to the gut epithelial surface and facilitating colonization [45]. By adhering tightly to host cells, STEC can resist phagocytosis by immune cells and avoid being cleared by the host’s innate immune defenses. In addition to this, STEC utilizes T3SS to inject a group of other effector proteins, such as EspF, Tir (translocated intimin receptor, Map (Mitochondrial-associated protein), EspG, EspG2, NleB (Non-LEE-encoded effector B), NleE, NleH, etc. (Table 2), into the host epithelial cells [57]. These effectors manipulate host cell processes, such as altering the actin cytoskeleton to form pedestal-like structures that secure bacteria more firmly to the cell surface. Additionally, T3SS effectors can suppress host immune responses, helping STEC evade detection and destruction by the immune system [45,57,58,59].

### 4.3. Damage and Evasion through Shiga Toxins (Stxs)

Stxs, such as Stx1 and Stx2, are potent cytotoxins prominently associated with pathogenesis in humans; however, they play a complex role in the colonization and adherence of STEC in the bovine gut. Stx toxins are AB5 toxins with a single A subunit and five B subunits. B subunits bind to globotriaosylceramide (Gb3) receptors on the surface of host cells and are internalized via receptor-mediated endocytosis [29,60]. Inside the cell, Stx undergoes retrograde transport from the Golgi apparatus to the endoplasmic reticulum (ER). In the ER, the A subunit, which demonstrates RNA N-glycosidase activity, cleaves a specific adenine base from the 28S rRNA of the 60S ribosomal subunit. This action inactivates ribosomes, halts protein synthesis, and leads to localized epithelial damage. It exposes deeper layers of the mucosa, providing new binding sites for STEC and enhancing its ability to persist within the host. Gb3 receptor distribution and expression level vary between humans and cattle. In humans, particularly children, Gb3 is highly expressed in the kidneys, which explains the prevalence of HUS following STEC infection [37]. However, cattle, natural reservoirs of STEC, have lower Gb3 expression levels in their intestines and typically do not develop HUS, allowing them to carry bacteria asymptomatically [3]. Additionally, the presence of Stx might modulate the host immune response, reducing the effectiveness of immune defenses and inflammatory responses, which allows bacteria to establish more robust colonization, as discussed in the previous section.

### 4.4. Tissue Penetration and Immune Evasion via Autotransporters

Autotransporters are also crucial in facilitating colonization of the bovine gut. Some notable autotransporters include OmpA, EspP, IcsA (VirG), Hbp, Pet, Ag43, Tsh, and EhaA-B-J (Table 1). The autotransporter protein EspP has proteolytic activity and degrades host proteins in the extracellular matrix and mucus layer, allowing STEC to penetrate deeper into the intestinal tissue. This degradation exposes some receptors on the epithelial surface, promoting STEC binding via other adherence factors, such as fimbriae and adhesins [61]. Another important autotransporter in STEC is the serine protease autotransporter of the Enterobacteriaceae (SPATE) family, which includes proteins Pet and Sat. These proteases can cleave host substrates, including mucins and immune components, weakening host defense mechanisms and promoting a more conducive environment for bacterial colonization [62]. Variability in the expression of autotransporters and host immune system recognition can lead to host-specific interactions. In bovines and all other hosts, biofilm formation by STEC is also crucial for the persistent colonization of the intestines and environmental survival. Autotransporters contribute to biofilm stability by providing structural components or mediating adhesion to surfaces or other cells [52]. In cattle, where host–pathogen interactions have co-evolved, the immune response to these autotransporters may be more controlled, often resulting in asymptomatic carriage rather than disease. For instance, cattle may express different receptor patterns or immune defenses that affect the outcome of STEC autotransporter activity compared to humans. Autotransporters may also influence the degree of STEC shedding in cattle, affecting the likelihood of its transmission to humans [3,63,64,65].

**Table 2 microorganisms-12-02009-t002:** Major STEC virulence factors and their role in bovine gut colonization.

Virulence Factor	Role in Bovine Gut Colonization	References
** *Shiga Toxins* **		
Stx1	Modulates the local immune response and contributes to the persistence of bacteria by dampening inflammatory signals and immune cell activation.	[12]
Stx2	Restricts epithelial cell proliferation within bovine crypts without inducing cell death and helps the bacteria maintain its niche within the gut, facilitating long-term colonization and shedding in cattle.	[12]
** *Adhesins* **		
Eae (Intimin)	Essential for strong adherence and colonization of the bovine gut epithelium by forming attaching and effacing (A/E) lesions.	[52]
Iha (IrgA homolog adhesin)	Supports adhesion and persistence under iron-limited conditions in the bovine gut.	[54]
Saa (STEC agglutinating adhesin)	Promotes strong adhesion to epithelial cells through autoagglutination, thereby facilitating the persistence and establishment of the bacteria within the host intestine.	[55,66]
Efa1/LifA (*E. coli* factor for adherence)	Enhances colonization by inhibiting the bovine immune response and aiding in bacterial adherence.	[65]
Type 1 fimbriae	Promotes initial attachment and colonization of the bovine gut by adhering to mannose-containing receptors on host cells.	[67]
F9 fimbriae	Contributes to colonization and persistence in the bovine gut by facilitating adhesion.	[3]
Long polar fimbriae (LPF)	Serves as a key factor in maintaining long-term colonization and persistence in the bovine gut.	[68]
Curli fimbriae	Enhances environmental persistence and colonization through biofilm formation in the bovine gut.	[69]
*E. coli* YcbQ laminin-binding fimbriae (ELF)	Supports colonization by adhering to laminin in the bovine extracellular matrix.	[70]
** *Autotransporters* **		
OmpA (Outer membrane protein A)	Modulates adherence of the bacteria to the rectoanal junction squamous epithelial cells, facilitating the interaction of other adhesins involved in the colonization process.	[71]
EspP (*E. coli*-secreted protein P)	Enhances colonization by degrading bovine host defenses through proteolytic activity.	[61]
EhaA-B-J	Contributes to stable colonization through adherence and biofilm formation in the bovine gut.	[52]
** *Type III Secretion System Effectors* **		
EspF	Disrupts tight junctions and alters the host cell cytoskeleton, which facilitates bacterial adherence and invasion and evasion of the host immune response, ultimately promoting bacterial survival and persistence within the bovine intestine.	[57]
Tir (Translocated intimin receptor)	Serves as a receptor for intimin, facilitating intimate adherence and the formation of attaching and effacing lesions in the bovine intestinal epithelium.	[3,45]
Map (Mitochondrial-associated protein)	Enhances bacterial survival and colonization by modulating host cell responses in the bovine gut.	[72]
EspG and EspG2	Disrupt the host cell microtubule network, leading to cytoskeletal alterations that enhance bacterial adherence and persistence.	[58]
NleB (Non-LEE-encoded effector B)	Is involved in initial adherence and plays a significant role in enhancing bacterial survival and persistence within the bovine gut by modulating host cell processes, such as the inhibition of apoptosis, thus promoting long-term colonization.	[45]
EspA, EspB, EspD	Form translocon pores and inject other effector proteins into the host cells, leading to the formation of attaching and effacing (A/E) lesions, which are essential for stable bacterial attachment and colonization in the bovine intestine.	[73]
NleH1 and NleH2 (Non-LEE-encoded effector1 and 2)	Inhibit the NF-κB signaling pathway, which dampens pro-inflammatory cytokine production, thereby creating a more favorable environment for STEC colonization and persistence.	[59]

## 5. Molecular Insights into STEC–Bovine Host Interactions

Recent molecular studies and advancements have significantly deepened our understanding of interactions between STEC and the host. These interactions are complex, involving several bacterial factors and host cellular processes. The use of transcriptomics, proteomics, and genomics has revealed the detailed mechanisms of STEC pathogenicity, offering new perspectives for mitigating STEC foodborne illness.

Transcriptomic analyses have revealed that STEC responds to the GIT environment by adjusting gene expression to enhance survival and colonization. For instance, studies have shown that STEC upregulates genes involved in iron acquisition, stress response, and toxin production upon entry into the host gastrointestinal tract. Molecular insights into the early stages of STEC colonization in cattle are critical to understanding and controlling its spread. A study compared transcriptomic profiles between nonpathogenic *E. coli* strain MG1655 and STEC (O26, O111, or O157) to identify the factors used by STEC during colonization of the bovine GIT. STEC demonstrated a numerically, though not significantly, higher level of adherence to cattle colonic explants compared to the nonpathogenic *E. coli* strain, suggesting a role of these upregulated STEC factors in early colonization. In particular, this study observed a significant upregulation of the flagellin gene (fliC) in O157 STEC and the Lon protease gene (lon) in all STEC strains when incubated with colonic explants. The upregulation of H7 fliC suggests that it serves as an adhesin in the bovine GIT. The collective upregulation of lon in STEC strains compared to that of nonpathogenic *E. coli* suggests its involvement in the stress response and, possibly, in regulating other virulence factors critical for colonization [74].

We recently examined variations in the gene expression of STEC O157:H7 when exposed to RAJ cells and human colonic epithelial cells (CCD CoN 841). This study aimed to understand how bacteria adjust their gene expression to adapt to the distinct environments of different hosts during initial attachment. STEC O157:H7 exhibited two different transcriptional profiles when attached to human and bovine intestinal cells, suggesting that STEC employs different strategies for colonization and survival in each host. In human cells, the enrichment of genes related to LPS polysaccharide and lipid biosynthesis indicates strategic manipulation by STEC to enhance adherence and possibly establish a niche within the host environment. STEC adapt to the host cell environment by regulating genes associated with metal ion homeostasis, which is crucial for its survival and colonization. Downregulation of pathways related to antibiotic resistance, drug metabolism, and secondary metabolism in bovine cells suggests that STEC prioritizes immediate colonization over secondary metabolic functions in cattle reservoirs. STEC upregulated genes involved in iron transport and utilization in human cells, highlighting the critical role of iron in bacterial growth and pathogenesis. This is in contrast to the downregulation of iron-related genes in bovine cells, indicating a different approach to iron usage in cattle. The study also identified the upregulation of specific virulence genes, such as those involved in heme utilization and the Type III secretion system (espW), demonstrating STEC’s pathogenic mechanisms of STEC during early host interactions. Overexpression of genes related to capsular polysaccharides and outer membrane proteins in the bovine host suggests strategies to evade immune responses and establish colonization. The upregulation of genes involved in colanic acid biosynthesis and extracellular polysaccharide production indicates the importance of biofilm formation for STEC survival and colonization in both human and bovine hosts. [75]. These findings elucidate the complex transcriptional adaptations of STEC O157:H7 in response to different host environments during initial attachment, providing insights into pathogen colonization strategies, virulence mechanisms, and potential targets for interventions to prevent STEC infection.

Like genomics studies, proteomics on STEC colonization in cattle also underscored the critical role of metabolic and phenotypic traits in determining adherence and colonization mechanisms of different STEC strains. Persistently colonizing STEC strains (STECper) and sporadically colonizing strains (STECspo) exhibit distinct metabolic profiles, particularly in their ability to utilize specific carbon and sulfur sources. Notably, the metabolisms of substrates such as glyoxylic acid and l-rhamnose were different between STECper and STECspo. Proteomic analyses have also identified mutations or disruptions in genes involved in the glyoxylate metabolism and rhamnose utilization pathways, which correlate with the colonization patterns observed in these strains. Additionally, the ability of STEC to form biofilms, a key proteomic trait, was found to be influenced by temperature, with STECspo producing more biofilms at lower temperatures as compared to STECper, suggesting an adaptation to environmental conditions [42]. These findings highlight the importance of metabolic versatility and ecological adaptability during STEC adherence and colonization of the bovine gut.

Genome sequence analysis of STEC strains isolated from cattle has provided insight into the genetic diversity, colonization factors, and evolution of STEC of bovine origin. Whole-genome sequencing and DNA microarray analysis have revealed that STECper isolates tend to possess a specific set of accessory genes absent in STECspo isolates. These accessory genes, including virulence-associated genes such as eae, nleA, nleB, and nleC, contribute to STEC adherence to the host intestinal epithelium, evasion of host immune response, and establishment of persistent infection. The presence of these genes and their variants correlates with the genetic background of STEC, as determined by multilocus sequence typing (MLST), with STECper clustering into separate phylogenetic groups. Moreover, the genomic plasticity of STEC allows for the horizontal transfer of these accessory genes, enabling the evolution of new virulent strains that can pose significant public health risks. These genomic differences between STECper and STECspo strains suggest formulating different mitigation strategies for different STEC genotypes to reduce STEC colonization of the bovine host [76].

Genome-wide association studies were conducted on a diverse collection of STEC isolates from various hosts, including cattle. These studies revealed that certain accessory genes were strongly associated with cattle-specific isolates, suggesting that these genes play a crucial role in the bovine host. Genes involved in iron acquisition, such as the iroBCDEN gene cluster, were found to be prevalent in cattle-associated *STEC*. These genes enhance the ability of bacteria to thrive in the bovine gut by effectively scavenging iron, a critical nutrient in the host environment. Additionally, the study identified variants of omptin proteins, such as OmpP, which are associated with cattle isolates and may contribute to the ability of bacteria to degrade antimicrobial peptides, thus aiding immune evasion and persistence within the bovine host [77]. These genomic insights underscore the complex interplay of genetic factors that enable STEC to adapt to and colonize the bovine gut, which might have implications for STEC mitigation approaches.

## 6. Bovine Gut Microbiota and STEC Colonization

The bovine gut microbiota also plays a crucial role in shaping the colonization dynamics of STEC [71]. The gut microbiota comprises a diverse community of microorganisms that coexist within the GIT, aiding host nutrition, metabolism, and immune system development. In the context of STEC colonization, the gut microbiota acts as both a barrier and a facilitator. On the one hand, a healthy and diverse gut microbiota can prevent pathogen colonization through competitive exclusion and the production of antimicrobial compounds [9,78,79]. Competitive exclusion refers to the mechanism by which resident microbiota outcompete potential pathogens for nutrients and attachment sites, thereby inhibiting colonization. Gut microbiota can also modulate the host immune response to enhance defense mechanisms against pathogens like STEC [79,80].

The role of gut microbiota in regulating STEC colonization and in several therapeutic strategies has been proposed. Administration of beneficial microbes (probiotics) or substrates that selectively stimulate their growth (prebiotics) can help restore or maintain a healthy gut microbiome, potentially reducing STEC colonization [79]. Specific probiotic strains (Lactobacillus plantarum and Lactobacillus fermentum) that inhibit STEC adhesion and toxin production have been identified [81]. Another potential intervention strategy, Fecal Microbiota Transplantation (FMT), which involves the transfer of fecal material from a healthy donor to the gastrointestinal tract of a recipient to restore a healthy microbial community, has also been explored [82]. This approach is particularly beneficial for managing recurrent or severe STEC infections because it aids in outcompeting pathogens by introducing healthy microbiota. Bacteriophage therapy utilizing bacteriophages that specifically target STEC can selectively reduce pathogen levels without disrupting the overall microbiota composition [83,84]. STEC-specific bacteriophages have both therapeutic and prophylactic uses against STEC colonization. Modification of host diet to promote the growth of beneficial microbes and inhibit the growth of pathogens can also be exploited to reduce STEC colonization of the bovine gut. In this context, dietary components that enhance the production of SCFAs or other antimicrobial metabolites by gut microbiota might be particularly valuable [85].

Apart from these approaches, additional methods for managing STEC in cattle include vaccination, nonmicrobial feed supplements, and monoclonal antibody-based treatments. Vaccination strategies targeting specific STEC components, such as Shiga toxins, have demonstrated promise in decreasing bacterial colonization and shedding [86]. Nonmicrobial feed additives such as organic acids and plant-derived antimicrobials can prevent STEC proliferation within the GIT [3]. The use of monoclonal antibodies that neutralize Stx has also been explored as both a preventive and therapeutic measure [86]. While these approaches are promising, this review emphasizes the approaches involving the gut microbiota because of their substantial role in preventing STEC colonization. A healthy and well-balanced gut microbiome is essential in impeding pathogen colonization through various mechanisms, such as competitive exclusion, the production of antimicrobial substances, and modulation of the host immune response. Consequently, this area presents a critical avenue for upcoming studies and intervention techniques.

## 7. Future Directions and Challenges

The study of host–pathogen interactions, mainly STEC colonization in bovine intestine, has advanced significantly. However, numerous gaps and challenges remain, necessitating a focused direction for future research. Although significant efforts have been made to understand STEC adherence and colonization mechanisms in cattle, the detailed strategies employed by STEC to adapt, survive, and thrive within the diverse environment of gut microbiota have not been fully elucidated. Understanding these adaptive mechanisms at a molecular level remains a critical challenge. The complex interplay between STEC, the gut microbiota, and the host immune system is not fully understood. Specifically, the role of non-bacterial components of the gut microbiota, such as fungi and viruses, in influencing STEC colonization and disease progression requires further exploration. Factors contributing to variability in bovine susceptibility to STEC colonization, including genetic, dietary, and environmental influences, require further investigation.

### 7.1. Challenges and Limitations

Studying STEC colonization in the bovine gut presents several challenges and limitations that complicate our understanding of pathogen behavior and the development of effective control strategies. One major challenge is the inherent genetic diversity among STEC strains [87], which leads to varying colonization patterns and virulence profiles, making it difficult to generalize findings across different strains. Additionally, the complex and dynamic nature of the bovine gut microbiome, which is influenced by factors such as diet, age, and environmental conditions, can affect the consistency and reproducibility of colonization studies. The use of animal models, although necessary for in vivo studies, also introduces limitations, as experimental conditions may not fully replicate the natural environment of commercial cattle herds. Furthermore, ethical considerations and logistical difficulties in conducting long-term studies on large animal populations can limit the scope and scale of research. Finally, it is challenging to translate findings from laboratory settings to practical on-farm interventions, as the controlled experimental conditions often do not account for the variability and complexity of the real-world farming environment. These challenges highlight the need for more refined methodologies and a multidisciplinary approach to better understand and mitigate STEC colonization in cattle.

### 7.2. Future Directions

Leveraging advanced omics technologies and bioinformatics tools to profile host–pathogen–microbiota interactions comprehensively is crucial. These approaches encompass single-cell sequencing, which can elucidate heterogeneity in the responses of STEC in human and bovine hosts during infection. Using CRISPR-Cas systems to dissect gene functions in STEC and the host is another approach to gain deeper insights into virulence mechanisms, host defense strategies, and potential therapeutic targets. Exploring novel prebiotics, probiotics, and dietary interventions to modulate gut microbiota composition could offer opportunities to mitigate STEC colonization and infection. A transdisciplinary approach to address gaps and provide new insights while overcoming these challenges will significantly advance our understanding of STEC colonization, host–pathogen interactions, and the role of gut microbiota in the bovine host. Ultimately, this knowledge will contribute to developing innovative and effective strategies to combat STEC infections and improve public health.

## Figures and Tables

**Figure 1 microorganisms-12-02009-f001:**
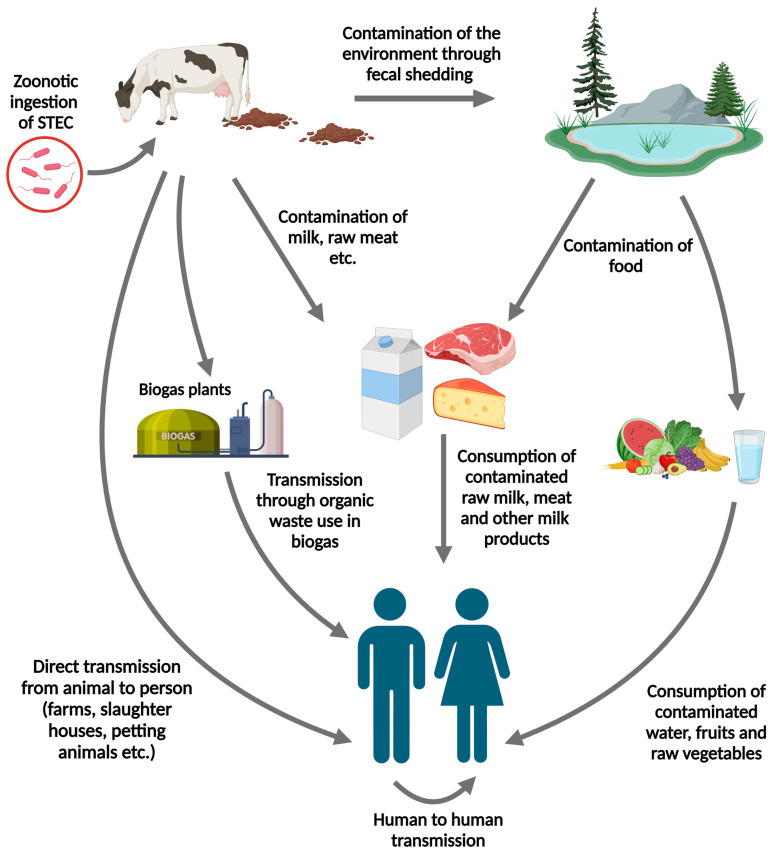
Mode of STEC transmission from cattle to humans (Created with BioRender.com).

**Table 1 microorganisms-12-02009-t001:** Overview of the prevalence of major STEC serogroups in the bovine gut.

STEC Serogroup	Prevalence in Cattle	Key Virulence Factors	Associated Diseases in Human	References
O157	High	Stx1, Stx2, Eae, HlyA	Hemorrhagic colitis, Hemolytic uremic syndrome (HUS)	[4,23,24,25]
O26	Moderate	Stx1, Stx2, Eae	Hemorrhagic colitis, HUS	[25,26,27]
O45	Low	Stx1, Stx2, Eae	Diarrhea, Hemorrhagic colitis	[28,29]
O103	High	Stx1, Eae	Diarrhea, Hemorrhagic colitis	[25,26]
O111	Moderate	Stx1, Stx2, Eae	Hemorrhagic colitis, HUS	[27,28,30]
O121	Low	Stx2, Eae	Diarrhea, Hemorrhagic colitis	[27,28,31]
O145	Low	Stx1, Stx2, Eae	Diarrhea, Hemorrhagic colitis	[26,29,32]

## Data Availability

No data were created or analyzed in this study.

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
