# Peer review of "Host–Pathogen Interactions during Shiga Toxin-Producing Escherichia coli Adherence and Colonization in the Bovine Gut: A Comprehensive Review"

_microorganisms, 2024, doi:10.3390/microorganisms12102009_

Round 1

Reviewer 1 Report

Comments and Suggestions for Authors

This interesting review submitted by Lekshmi K. Edison, Indira T Kudva, Subhashini and Kariyawasam elegantly discusses the problem related to Shiga toxin-producing Escherichia coli (STEC), which is characterized as a major public health threat due to its ability to cause severe gastrointestinal diseases in humans, ranging from diarrhea to fatal conditions such as hemorrhagic colitis and hemolytic uremic syndrome (HUS).

The initial reservoir of this microorganism is cattle, and this animal nutrient source plays a relevant role in its transmission through contaminated food and water, posing a considerable risk to human health.

The importance of this comprehensive review lies in the fact that it explores in detail the host-pathogen interactions during STEC colonization of the bovine intestine, with a focus on the role of the intestinal microbiota in modulating these interactions and influencing disease outcomes.

The authors were able to discuss important findings from transcriptomic, proteomic and genomic studies that have been published by different research groups to provide a complete understanding of how STEC adheres to and colonizes the bovine gastrointestinal tract. The information contained in this review describes new potential horizons for the development of new preventive and therapeutic strategies aimed at controlling STEC colonization in cattle, thus reducing the risk of zoonotic diseases. In short, an interesting and relevant review for the field of study.

Author Response

Comment 1: This interesting review submitted by Lekshmi K. Edison, Indira T Kudva, Subhashini and Kariyawasam elegantly discusses the problem related to Shiga toxin-producing Escherichia coli (STEC), which is characterized as a major public health threat due to its ability to cause severe gastrointestinal diseases in humans, ranging from diarrhea to fatal conditions such as hemorrhagic colitis and hemolytic uremic syndrome (HUS).

The initial reservoir of this microorganism is cattle, and this animal nutrient source plays a relevant role in its transmission through contaminated food and water, posing a considerable risk to human health.

The importance of this comprehensive review lies in the fact that it explores in detail the host-pathogen interactions during STEC colonization of the bovine intestine, with a focus on the role of the intestinal microbiota in modulating these interactions and influencing disease outcomes.

The authors were able to discuss important findings from transcriptomic, proteomic and genomic studies that have been published by different research groups to provide a complete understanding of how STEC adheres to and colonizes the bovine gastrointestinal tract. The information contained in this review describes new potential horizons for the development of new preventive and therapeutic strategies aimed at controlling STEC colonization in cattle, thus reducing the risk of zoonotic diseases. In short, an interesting and relevant review for the field of study.

Response 1: Thank you for your thoughtful and positive feedback on our review.

Reviewer 2 Report

Comments and Suggestions for Authors

This paper is devoted to the interaction of pathogenic shiga toxin-producing Escherichia coli (STEC) with the bovine. The serotypes of the pathogen, its interaction with the host at the cellular, biochemical and genetic levels are comprehensively considered. The work is of undoubted scientific interest for specialists and should be accepted for publication in the journal. However, it is necessary to make minor adjustments and additions that do not affect the quality of the work.

Item 3.1. The authors draw attention to the relationship between acidity in different parts of the gastrointestinal tract and the growth of STEC. This part of the work is given in a blurry form. Please describe in more detail how pH affects the growth of STEC bacteria and how pH changes in different parts of the gastrointestinal tract of bovines, including areas favorable for colonization by STEC.

Item 3.2. Is there any evidence on the effect of STEC-induced immune suppression on cattle susceptibility to other infectious diseases? If so, could you briefly describe it?

Item 6. In this item, the authors describe the control of STEC bacteria using probiotics, microbiota transplantation methods, and bacteriophages. Are there other methods of treating and preventing STEC in bovines? If so, please describe them in a separate item.

Author Response

Comment 1: This paper is devoted to the interaction of pathogenic shiga toxin-producing Escherichia coli (STEC) with the bovine. The serotypes of the pathogen, its interaction with the host at the cellular, biochemical and genetic levels are comprehensively considered. The work is of undoubted scientific interest for specialists and should be accepted for publication in the journal. However, it is necessary to make minor adjustments and additions that do not affect the quality of the work.

Response 1: Thank you for your thoughtful and positive feedback

Comment 2: Item 3.1. The authors draw attention to the relationship between acidity in different parts of the gastrointestinal tract and the growth of STEC. This part of the work is given in a blurry form. Please describe in more detail how pH affects the growth of STEC bacteria and how pH changes in different parts of the gastrointestinal tract of bovines, including areas favorable for colonization by STEC.

Response 2: Thank you for your valuable comment. While there is limited literature on pH variations in the bovine gastrointestinal tract and STEC adaptation, we have thoroughly detailed the available data in our description as follows:

 “The pH levels throughout the bovine GIT create favorable regions for the survival and proliferation of STEC. The rumen has a relatively neutral pH, favorable to diverse microbiota, and serves as an important reservoir for STEC. However, as the digesta containing STEC moves through the GIT, the pH levels fluctuate, creating varying acidic conditions, particularly in the abomasum and small intestine, before STEC stabilizes in the large intestine. These fluctuations in pH provide niches that STEC can uniquely exploit because of its ability to tolerate a wide range of pH (43). Unlike many other bacteria that cannot withstand lower pH levels, STEC is well adapted to survive in relatively high acidic environments and effectively colonize downstream to the rumen, especially the colon. This acid resistance allows STEC to bypass harsh conditions in the stomach and effectively colonize regions such as the colon, where the pH is higher and more stable. STEC can attach to the intestinal epithelium in the colon, establishing colonization sites and enabling persistence and potential transmission through fecal shedding (44)”.

Comment 3: Item 3.2. Is there any evidence on the effect of STEC-induced immune suppression on cattle susceptibility to other infectious diseases? If so, could you briefly describe it?

Response 3: Thank you for raising this important point. After a thorough review of the available literature, we could not find any direct reports addressing STEC-induced immune suppression's effect on cattle's susceptibility to other infectious diseases. While some studies explore the immune-modulating effects of Shiga toxins in general, none provide conclusive evidence regarding how these effects might increase vulnerability to other infections in cattle.

Comment 4: Item 6. In this item, the authors describe the control of STEC bacteria using probiotics, microbiota transplantation methods, and bacteriophages. Are there other methods of treating and preventing STEC in bovines? If so, please describe them in a separate item.

Response 4: Thank you for your question. In addition to probiotics, microbiota transplantation, and bacteriophages, several other approaches have been explored for the treatment and prevention of STEC infection in bovines. These include vaccination strategies targeting specific STEC components, using non-microbial feed additives, dietary management techniques, and administering monoclonal antibodies against STEC toxins. We have included this in a separate paragraph of Section 6.

“Apart from these approaches, additional methods for managing STEC in cattle include vaccination, non-microbial feed supplements, and monoclonal antibody-based treatments. Vaccination strategies targeting specific STEC components, such as Shiga toxins, have demonstrated promise in decreasing bacterial colonization and shedding (88). Non-microbial feed additives such as organic acids and plant-derived antimicrobials can prevent STEC proliferation within the GIT (3). The use of monoclonal antibodies that neutralize Stx has also been explored as both a preventive and therapeutic measure (88). While these approaches are promising, this review emphasizes the approaches involving the gut microbiota because of their substantial role in preventing STEC colonization. A healthy and well-balanced gut microbiome is essential in impeding pathogen colonization through various mechanisms, such as competitive exclusion, the production of antimicrobial substances, and modulation of the host immune response. Consequently, this area presents a critical avenue for upcoming studies and intervention techniques.”